# Resilience in the Storm: Impacts of Changed Daily Lifestyles on Mental Health in Persons with Chronic Illnesses under the COVID-19 Pandemic

**DOI:** 10.3390/ijerph18115875

**Published:** 2021-05-30

**Authors:** Bobo H. P. Lau, Mike K. T. Cheung, Lucian T. H. Chan, Cecilia L. W. Chan, Pamela P. Y. Leung

**Affiliations:** 1Department of Counselling and Psychology, Hong Kong Shue Yan University, Hong Kong, China; 2Hong Kong Society for Rehabilitation, Hong Kong, China; cthl1225@gmail.com (L.T.H.C.); pamela.leung@rehabsociety.org.hk (P.P.Y.L.); 3Department of Social Work and Social Administration, Center on Behavioral Health, The University of Hong Kong, Hong Kong, China; cecichan@hku.hk

**Keywords:** chronic illness, resilience, mental health, psychological adjustment, COVID-19, SARS-CoV-2, disparities

## Abstract

Studies have shown individuals with chronic illnesses tend to experience poorer mental health compared to their counterparts without a chronic illness under the COVID-19 pandemic. The pervasive disruption on daily lifestyles due to social distancing could be a contributing factor. In this study, we collaborated with local patient support groups to explore the psychological adjustment among a group of community-dwelling individuals with chronic illnesses under the COVID-19 pandemic in Hong Kong. We collected responses from 408 adults with one or more chronic illnesses using an online survey. Results show that about one in four participants experienced moderate to high levels of depression (26.0%), anxiety (26.2%) and stress (20.1%) symptoms measured by the Depression, Anxiety and Stress Scale and the World Health Organisation-Five Well-Being Index. While 62.3% (gatherings) to 91.9% (contact with others) of participants reported changes in their daily lifestyles, these changes—both an increase and a decrease—were related to poorer mental health. The relationship was mediated by psychological resilience, measured by the Connor–Davidson Resilience Scale, with an estimate of indirect effect of −0.28 (95% confidence interval −0.44 to −0.10). In light of our findings, we urge social and healthcare professionals to support chronic illness patients to continue their daily lifestyles such as exercises and social contacts as much as possible by educating the public on feasible and practical preventive measures and enhance the psychological resilience of community-dwelling patients with scalable and efficacious psychological interventions.

## 1. Introduction

The coronavirus disease (COVID-19) pandemic has led to a global mental health crisis. The novelty of the virus, the uncertainty of its course and long-term health impacts, the unknown prospect of mass vaccination and pandemic control policies and the severe consequences on the economy and society have generated an unprecedented challenge to the global community. Unlike other catastrophes, the extensiveness and pervasiveness of the COVID-19 pandemic is expected to leave behind a long trail of loss-related distress (e.g., loss of loved ones, health, job, social networks, daily routine) and intense worries and anxiety (e.g., unknown arrival of another outbreak) [1]. A review of 19 studies published up to May 2020 has found high rates of anxiety (6.33% to 50.0%), depression (14.6% to 48.3%), psychological distress (34.4% to 38%) and stress (8.1–81.9%) among samples of general populations in Asia, North America and Europe [2]. McKinsey & Company have estimated that in the U.S. alone an additional 35 million individuals with mental health needs will emerge, with 1.6 million due to the direct exposure to COVID-19 illness and loss and the rest due to the impacts from changed lifestyles, stretched healthcare provision, economic downturn, etc. [3].

The COVID-19 pandemic has presented distinct challenges to the population with chronic illnesses such as diabetes, chronic respiratory diseases, cardiovascular diseases, cancer or compromised immunity. Heightened anxiety over an infection and the risk of mortality as well as the drastic changes to their daily lifestyles may have rendered them vulnerable to greater psychological distress. This study therefore investigated how the disruption to daily lifestyles may influence psychological resilience and mental health among the community-dwelling population with chronic illnesses in Hong Kong.

### 1.1. Pandemic Adjustment for Persons with a Chronic Illness

In Hong Kong, 20.8% of the population suffer from chronic illnesses including high blood pressure, diabetes, chronic heart disease, cancer, asthma and stroke [4]. Even pre-pandemic, emotional distress and social isolation tended to be more common among persons with these conditions compared to their healthy peers [5]. During the COVID-19 pandemic, studies have found persons with chronic illnesses fared worse in terms of psychological distress [6,7,8], fear of COVID-19 [9] and infection-related worries [10] than those without a health condition. It is noteworthy that these studies regarded persons with a chronic illness as those with a physical illness but not a psychiatric condition. Heightened worries over an infection, a more severe course of illness and higher odds of mortality may elicit more avoidant or hypervigilant behaviors. In turn, the upended daily lifestyles (e.g., increase in sedentary time, reduced activity levels) and disruption to routine health care (e.g., cancellation of clinic visits, reduced medical adherence, delayed medical procedures) may impose further short or long-term health consequences [3,11]. Smith and colleagues articulated the exhausting struggles of persons with chronic illnesses to comply with the well-intended social distancing recommendations and to cope with the intensified social isolation, loneliness and psycho-socio-economic sequalae with the notion known as the ‘COVID Social Connectivity Paradox’ [12]. This paradox highlights the ‘double-edged sword’ nature of social distancing that generates protection against an infection at the expense of escalated psychosocial distress in vulnerable populations. The idea underscores the need for interventions that balance these contradicting objectives in order to preserve social connections and salutary routines in spite of the pandemic.

### 1.2. Daily Lifestyle Disruption, Psychological Resilience and Mental Health

The drastic changes in daily routines due to the pandemic control policies, such as the suspension of school and work, cancellation of regular gatherings (e.g., volunteering, church, outdoor hobbies), swapping offline contacts with online modes and the sharp reduction of outdoor activities may impair one’s mental health. For instance, Giuntella and colleagues reported a marked decrease in physical activity and an increase in screen time and sleep duration among a sample of American university students comparing the spring of 2019 to that of 2020 [13]. While the disrupted routines were related to increased depression in the sample, a simple intervention using Fitbit (rewarding $5 per day for achieving 10,000 steps) restored physical activity to the pre-pandemic level but failed to ameliorate the psychological distress. Likewise, a study on community-dwelling seniors in Italy reported a drastic reduction in physical activity, adherence to the Mediterranean diet, social activities and cognitively stimulating activities as well as an increase in idle time during the height of the local outbreak [14]. The reduction in productive activities was in particular related to poorer mental health in the sample.

Psychological resilience refers to the perceived ability of a person to cope with and bounce back from an adversity and could be a mediator of successful coping [15,16,17]. In the development of the Connor–Davidson Resilience Scale (CD-RIS), the authors remarked they operationalized the construct by referring to the earlier works of the following: Kobasa on hardiness which encompassed control, commitment and viewing a change as a challenge [18]; Rutter’s work which emphasized secure relationships, self-efficacy, past success, perceived choice and action orientation as protective factors against psychiatric disorders [19]; and Lyon’s work on patience and distress tolerance for adjusting to a trauma [20]. During the COVID-19 pandemic, several studies have reported negative relationships between psychological resilience and sense of danger, and psychological distress and somatic symptoms in community samples [21,22,23]. Psychological resilience was also found to mediate the link between a pandemic-related stressful experience and acute stress disorder among a sample of Chinese university students [24].

However, the upended daily routines may jeopardize psychological resilience. The Drive to Thrive Theory (DTT) [25] asserts that resilience is determined by whether the routines and structures of one’s daily life can be sustained. In stressful times, people may struggle to sustain their everyday practices (e.g., rest, diet, exercise, work, entertainment). When the damage to their everyday routines escalates beyond a particular ‘breaking point’, a rapid breakdown in lifestyle structures is induced, resulting in a quick deterioration in health outcomes. In other words, resilience is supported by the daily routines and structures that resist the shock and damage caused by a stressor. Although the DTT conceptualizes resilience as a dynamic coping trajectory characterized by a sustained absence of marked distress in spite of the stressor, the lesson on the importance of the integrity of daily routines on people’s coping resources may offer a fruitful perspective to understanding why the COVID-19 pandemic is so hurtful to one’s mental health, especially among those with a chronic illness. Accordingly, Killgore, Taylor, Cloonan and Dailey revealed a decrease in psychological resilience among a sample of American youths during the first weeks of lockdown in the spring of 2020 [26]. They also observed the levels of resilience being correlated with time spent outdoors, perceived social support, frequency of prayers and duration of daily exercises, in addition to psychological distress.

### 1.3. The Current Study

In Hong Kong, the first case of COVID-19 was confirmed on 23 January 2020. As of May 2021, the city has had more than 11,800 cases with over 200 deaths. During the local outbreaks in 2020, the government relied on several strategies to contain the spread, including mass testing with the ambush lockdown of blocks with suspected cases, banning public gatherings, limiting dine-in services, mandatory closure of high-risk premises including bars, saunas and sports venues, school closures and suspension of non-essential public services. A territory-wide lockdown did not happen up to the time of this writing in May 2021.

Hong Kong was the ‘ground zero’ of SARS in 2003. With the memory of SARS still fresh for most adults, since the beginning of the COVID-19 pandemic the public has shown a heightened awareness and readiness to act by stocking up on face masks and alcohol hand sanitizers, cancelling gatherings, limiting cross-border and international travels, etc. [27,28]. Nonetheless, the pandemic turned out to have lasted much longer than SARS and entailed greater impacts to the daily lives of the citizens, with a severe economic downturn and the largest increase in the unemployment rate since 2003.

This study was conducted to explore the psychological adjustment among a group of community-dwelling individuals with chronic illnesses under the COVID-19 pandemic. The data were collected between 1 October and 15 November 2020, where about 10 confirmed cases of COVID-19 were reported daily and the mass vaccination program had yet to begin. The period was between two outbreaks: one in late July/August, and the other one beginning in late November in the same year. The later outbreak had been largely forewarned by health experts and anticipated by the public. Hence, even though some social distancing measures (e.g., closure of high-risk venues, suspension of public services and school) had been relaxed, the public remained generally vigilant. The use of face masks in public areas remained mandatory and gathering bans and a limitation on dine-in services to four persons were still in place. Therefore, most social gatherings (e.g., church worship, weddings, volunteer meetings, training courses) remained suspended or had to drastically reduce the number of attendees.

We tested whether psychological resilience mediates the link between changes in daily lifestyles and mental health. We measured perceived changes in one’s physical exercises, outdoor activities, utilization of services of non-government organizations (NGOs), utilization of services of self-help organizations (SHOs), contact with others and social gatherings and investigated whether changes in these aspects are related to psychological resilience and mental health. Although we expected that most participants would report a decrease in these activities due to social distancing, there could be a small number of participants reporting an increase due to escalated care duties to another vulnerable person, or the nature of their occupation. As a decrease in these activities might have led to boredom and inconvenience, while an increase would have exposed oneself to additional risks of infection, it is possible that a change, either increase or decrease, would have challenged the daily routines of the participants in such a difficult time, and therefore hampered their psychological resilience and mental health.

## 2. Materials and Methods

### 2.1. Participants

Participants of this survey were adults aged 18 or above, able to understand Chinese and self-reported with at least one chronic illness. They were recruited from community centers as well as self-help organizations for people with chronic illnesses, and via online marketing channels including email lists of support groups, Facebook posts, and instant messages (e.g., WhatsApp). The advertisement described the study as one that explores local adults’ psychological adjustment in the COVID-19 pandemic and explicitly asked for adults with chronic illnesses to participate.

### 2.2. Procedures

The study was conducted through an online survey hosted on SurveyMonkey which took about 20 min to complete. The survey included questionnaires on mental health, psychological resilience, daily lifestyles and demographic information. Participants provided their informed consent before the beginning of the survey. Participation in the survey was anonymized and participants were not monetarily reimbursed after their participation. The study was approved by the Human Research Ethics Committee of the Hong Kong Shue Yan University (approval no.: HREC 20-09 (7)).

### 2.3. Measurements

Mental Health: The 21-item Chinese version of the Depression, Anxiety and Stress Scale (DASS-21) [29] and the Chinese version of the World Health Organisation- Five Well-Being Index (WHO-5) [30] were adopted to measure participants’ mental health. These two instruments were both used in other local studies investigating the mental health status of the general public [31,32]. DASS-21 yielded three scores that ranged from 0 to 21 for depression, anxiety and stress and the participants were asked to respond on a scale running from 0 (not at all) to 3 (almost always). Higher scores on the subscales indicated more distress with cutoff scores of 6/7, 5/6 and 9/10 indicating moderate or more depression, anxiety and stress. WHO-5 contained 5 items responded to on a scale running from 0 (at no time) to 5 (all of the time). The WHO-5 yielded a single score ranging from 0 (the worst imaginable well-being) to 100 (the best imaginable well-being) with a cut-off score of 50 indicating the absence of depression [33]. These two instruments showed good internal consistency in this study (Cronbach’s αs ranged from 0.73 to 0.92).

Psychological resilience: The 10-item Chinese version of the Connor–Davidson Resilience Scale (CD-RIS-10) was adopted [34]. The ten items were responded to on a scale running from 0 (not true at all) to 4 (true nearly all of the time) with higher scores indicating higher psychological resilience. CD-RIS-10 showed good internal consistency in this study (Cronbach’s α = 0.94).

*Changes in daily lifestyles:* A set of indicators were developed to measure participants’ self-reported changes in daily lifestyles due to the COVID-19 pandemic including physical exercises, outdoor activities, utilization of services of non-government organizations (NGOs), utilization of services of self-help organizations (SHOs), contact with others and social gatherings. Participants were asked if they have experienced any change in these aspects (decrease, increase, the same, no such activity even pre-pandemic) since the local outbreak of COVID-19. The indicators were selected according to the common and critical factors for individuals with chronic illnesses to maintain their physical and mental health after consulting various patient support groups and being tested by service clients with chronic illnesses to ensure face validity.

Demographic and clinical characteristics including age group, gender, types of chronic illnesses and occupational status were also collected. The disabling score of the chronic illnesses was derived according to the scoring guideline proposed by Cournane et al. (2015) to account for the effect of comorbidity [35]. This disabling score counted the number of the eight systems/disabling categories (including cardiovascular, respiratory, neurological, gastrointestinal, diabetes, renal, neoplasms, other). Participants with a diagnosis in one of these eight categories were given a score of 1. Hence, the disabling score ranged from 0 to 8.

### 2.4. Statistical Analysis

Descriptive statistics were used to explore participants’ demographic and clinical characteristics and the key variables included in the mediation model. The relationships among the changes in daily lifestyles, mental health and psychological resilience were explored with the following: bivariate correlations (Pearson correlation coefficient) for two continuous variables (e.g., outcome variables), independent sample t-test (for dichotomous variables or not) and continuous variables (e.g., outcome variables), and analyses of variance (ANOVAs) with post-hoc tests with Tukey’s test for categorical variables (e.g., changes in daily lifestyle) and continuous variables (e.g., outcome variables). Structural equation modeling (SEM) was used to examine the hypothesized mediation between changes in daily lifestyles and mental health via psychological resilience. A latent variable was constructed by the six indicators of daily lifestyle changes (physical exercises, outdoor activities, utilization of services of NGOs, utilization of services of SHOs, contact with others and social gatherings). Another latent variable, mental health, was constructed by the three scale scores of DASS-21 (depression, anxiety and stress) and WHO-5. Since DASS-21 measures emotional distress while the WHO-5 indicates mental well-being, the three scale scores of DASS-21 were reversed in the model to bring them in line with the interpretation of WHO-5. The scale score of CD-RIS-10 was treated as an observed variable and included as the mediator in the SEM. Age, gender and the disabling score of chronic illnesses were adjusted in the model. Multiple indicators were used to indicate the model-data fit of the SEM model, including the Goodness of Fit Index (GFI ≥ 0.9), Root Mean Square Error of Approximation (RMSEA < 0.08) and Standardized Root Mean Square Residual (SRMR ≤ 0.08) [36]. The bootstrapping of the repeated 10,000 sample was used to yield pairs of 95% confidence intervals for evaluating the statistical significance of the direct and indirect effects at an alpha of 0.05. All statistical analyses were performed by R and the SEM analysis was performed by the Lavvan package of R.

## 3. Results

Four hundred and eight participants completed the online survey and were included in the analysis. The sample characteristics are provided in Table 1. Around one third of participants were aged 65 (38.5%) or above and between 55 and 64 (35.5%), while 26.0% were aged below 55. More than half of them were female (61.3%). The most common chronic illnesses were hypertension (40.9%), diabetes (30.1%) and heart disease (15.0%). More than half of the participants had only one disabling chronic condition (59.6%). Around 30% of them were working full-time, part-time or were self-employed. The mean of participants’ perceived risk of infection of COVID-19 (score ranged from 0 to 10, higher score indicates higher perceived risk) was 3.79 (SD = 2.12), which was relatively low. The majority of respondents (94.9%) had experienced the SARS epidemic in Hong Kong in 2003.

Table 2 provides the descriptive statistics and intercorrelations of the outcome variables. The mean scores of depression, anxiety and stress of DASS-21 were 7.64 (SD = 8.47), 8.02 (SD = 7.35) and 10.93 (SD = 9.08), respectively. Furthermore, 26.0%, 26.2% and 20.1% of participants showed moderate or higher levels of depression, anxiety and stress, respectively. The mean score of WHO-5 was 47.66 (SD = 22.44) and 52.7% of participants showed mild or more depressive symptoms. The mean score of CD-RIS-10 was 26.99 (SD = 7.86). The indicators of mental health and psychological resilience were moderately to strongly correlated (|r|s > 0.40) in expected directions. Participants with a higher age showed a lower score of depression, anxiety and stress of DASS-21 and a higher score of WHO-5 and CD-RIS-10. Gender difference in the scores of DASS-21, WHO-5 and CD-RIS-10 were not significant, except anxiety where the female participants reported higher scores than the male participants (see Appendix A for detailed results).

Table 3 provides the frequencies of changes in daily lifestyles. More than 80% of participants reported changes in their daily lifestyles or none of these activities including gathering with others (91.9%), utilization of SHO services (89.7%) and utilization of NGO services (87.0%). More than half of the sample also reported changes in contact with others (62.3%), physical exercise (65.0%) and outdoor activities (74.0%). As expected, under social distancing recommendations, most participants reported a decrease (ranging from 46.3% to 89.0%), rather than an increase in these activities. Only a minority reported they did not have these activities even pre-pandemic.

### 3.1. Relationships between Changes in Daily Lifestyles and Psychological Resilience and Mental Health

Results of the ANOVAs with post-hoc analysis demonstrate that among the 18 out of 30 (5 outcome indicators × 6 change in daily lifestyles) significant omnibus effects on psychological resilience and mental health (see Appendix A for detailed results), 15 were accompanied by a significant post-hoc comparison between the group who experienced a reduction or increase in these activities or did not have these activities pre-pandemic versus those who kept the same amount of activities. Only three post-hoc comparisons were significant with respect to the increase versus the decrease of activities. As expected, a change in the frequencies of these activities is what matters to psychological resilience and mental health, rather than an increase or a decrease per se. Therefore, in the subsequent analyses, we grouped participants who reported an increase or a decrease or not having the activities pre-pandemic into one group; whereas those who reported consistent frequencies were grouped in another. The comparisons of the psychological resilience and mental health between these two groups are presented in Table 4. Participants reporting either a change in their daily lifestyle activities or not having the activities pre-pandemic showed poorer psychological resilience and mental health in general, compared to their peers who reported no change in the frequencies of the activities. Nineteen of the 30 comparisons (63.3%) reached statistical significance (*p* < 0.05).

### 3.2. Mediation Effect of Psychological Resilience

The latent variable, changes in daily lifestyles, was indicated by six binary items (change/none pre-pandemic coded as 1, no-change coded as 0). The model-data fit was acceptable (GFI = 0.91, RMSEA = 0.09 and SRMR = 0.08) (see Figure 1 for the path coefficients). A post-hoc power analysis showed the sample offers 95.1% of statistical power to detect model misspecification in the SEM analysis. All observed variables of mental health and changes in daily lifestyles were significantly regressed on their corresponding latent variables. Changes in daily lifestyles were significantly related to psychological resilience (β = −0.31, *p* < 0.01) but not mental health (β = −0.15). Psychological resilience was significantly related to mental health (β= 0.90, *p* < 0.001). The total effect from changes in daily lifestyles to mental health was −0.43 while the direct effect (−0.15) accounted for 35.2% of the total effect but was non-significant (Table 5). The indirect effect via psychological resilience was −0.28 and accounted for 64.8% of the total effect. The findings support the mediation via psychological resilience in the relationship between changes in daily lifestyles and mental health. More changes in daily lifestyles were related to poorer psychological resilience, and in turn worse mental health outcomes.

## 4. Discussion

This study was conducted in collaboration with a network of patient support groups in Hong Kong in order to examine the psychological adjustment of the population with chronic illnesses under the COVID-19 pandemic. Our findings show that about one in four to five participants experienced moderate or more depression, anxiety or stress symptoms. The rates were comparable to other local studies (14.0% to 33.9%) [28,37,38]. However, our data were collected during a period (October to November 2020) between two major outbreaks. Outbreak severity, indicated by the incidence of cases or mortality, tends to be positively related to emotional distress [39]. Hence, we consider our rates as valid yet alarming as they remained high despite being collected during a time with relatively less COVID-19 cases. Compared to the population quartile scores derived from a community sample of American adults [40], our sample mean was close to the 25th percentile (score = 29), meaning that our participants tended to report relatively low psychological resilience. While the COVID-19 prevention policies are well-intended, the ‘new normal’ requires a great deal of adjustment, psychosocially and practically, among the population with chronic illnesses, thus challenging their psychological resilience. Of note: our data were collected almost eight months after the first case of COVID-19 in Hong Kong. Hence, our findings may indicate the sustained difficulties of adjustment among this vulnerable population.

As hypothesized, our findings show that changes in daily lifestyles—no matter an increase or a decrease in outings, social gatherings or usage of support services—were related to poorer psychological resilience and mental health. During the COVID-19 pandemic, a decrease in social contacts or outings may lead to boredom, feelings of isolation or practical inconvenience; however, an increase in these activities may expose one to additional infection risk. These increases in social activities and outings could be due to escalated care needs of another vulnerable person at home, or work requirements. It is doubtful whether adequate protection has been available for these individuals to counteract the heightened infection risk. Understanding that personal protection equipment (e.g., a good ventilator, multiple face masks per day) could be costly or inadequate in various occupational contexts, some of these contacts and outings could be considered risky and undesirable by our participants. As the COVID Social Connectivity Paradox [12] highlighted, adherence to social distancing has become a difficult struggle for vulnerable groups as it requires balancing the risk, benefits and costs. A study conducted in Mexico City also found elderly persons with lower income and education tended to underestimate the severity of the pandemic and susceptibility to the infection, and therefore reported lower adherence to preventive behaviors [41]. Hence, the first set of suggestions we have for social and health care professionals is to issue practical, up-to-date and feasible guidelines for the population with chronic illnesses to minimize their infection risk while maintaining their healthy physical and social daily routines.

Changes in socializing and outings were found to hamper psychological resilience for adults with chronic illnesses. In other words, our findings replicated those of Killgore et al. [26]. It could be wrong to assume that the population with chronic illnesses, with lower baseline socializing and outings, are unaffected by social distancing. In fact, participants of our study who reported not having these activities pre-pandemic also fared worse than their counterparts who have sustained their activities. Our findings also postulated that psychological resilience, as a construct amenable to psychological interventions, mediates the inevitable yet detrimental effects of daily lifestyle disruptions. As the COVID-19 pandemic is likely to continue affecting the daily routines of various vulnerable populations (e.g., the elderly, persons with chronic physical illnesses and their caretakers) until herd immunity is reached through global mass vaccination programs, we suggest social and healthcare professionals develop and provide scalable, efficacious psychological support through tele-medicine or tele-counselling. It is foreseeable that the legacy of such online counselling efforts will also benefit populations with difficulty accessing conventional, physical means even post-pandemic. A meta-analysis with eleven randomized controlled trials found that mindfulness and/or cognitive behavioral therapy-based interventions are effective for improving psychological resilience compared to control conditions, with some evidence of sustainable benefits up to six months post-intervention [42]. Another meta-analysis reported that stress management, positive psychology, group coaching, disease education, Tai Chi and relaxation may improve psychological resilience among individuals with chronic illnesses [43]. While some of these interventions could be adapted to an online mode through smartphone applications, attrition rates could remain an issue. Gamification, offering monetary compensation, regular reminders and online (rather than offline) enrollment may foster retention of smartphone delivered programs [44,45].

This study is distinctive in two ways. First, it has been collected from a location heavily hit by SARS in 2003 and without a territory-wide lockdown. Hence, it reveals a resilient community scenario under the COVID-19 pandemic. Second, it has targeted persons with chronic illnesses, who are regarded as a vulnerable group due to their higher risk of infection and more severe disease course in case of infection. In fact, the demographic characteristics of our sample (e.g., occupation status, age range) were largely similar to those reported on the thematic household survey on persons with chronic illnesses conducted by the Census and Statistics Department in 2019 [4], which lends support to the external validity of our findings.

However, several limitations are note-worthy. Some sub-populations including males, people with limited access to the internet or those with little contact with patient support networks could have been underrepresented. As people without access to the internet or support groups may also lack access to accurate pandemic-related information and resources, they could be in a more dire situation, rendering our findings an overly optimistic estimate of the mental distress of the population. Wong et al. [46] reported that perceived benefits and harms of the pandemic on one’s family well-being were unevenly distributed across demographic groups in Hong Kong, with males and people with lower socio-economic status reporting more adverse impacts. Hence, we call for future studies to utilize representative datasets to study the positive and negative sequalae of the pandemic. In addition, this study has been conducted as a swift response to the changing circumstances of the pandemic and therefore relied on a non-random sample. The survey’s response rate was unavailable as we relied on a diverse set of networks to maximize the reach of the survey among chronic illness patients in Hong Kong. Furthermore, the cross-sectional nature of the study did not permit inferences regarding the direction of causality. Although the DTT [25] conceptualizes the breakdown in daily routines and structures as the precursor of poor psychological resilience and mental health, future studies may utilize longitudinal designs to explore how poor psychological resilience and mental health may impede the restoration of daily lives after the pandemic. Furthermore, although most community-dwelling individuals with chronic illnesses share a similar concern for heightened infection risk, there is considerable heterogeneity with respect to the needs arising from different diagnoses and medical treatments (e.g., therapies that compromise immunity). We also lacked data on participants’ medical history, experience of stigma from engaging in the healthcare industry or from having a family member or cohabitant test positive on SAR-CoV-2 and the use of applications for psychological self-care and were therefore unable to account for the impacts of these factors. Lastly, our assessment of the changes in daily lifestyles was based on the opinions of local patient support groups. Thus, it may not have exhausted the daily life concerns of patients in other socio-cultural contexts.

Mass vaccination programs have been rolled out globally. Many countries have given priority access of vaccines to elderly persons and persons with chronic illnesses. Studies have documented more positive attitudes to vaccines and higher intention to vaccinate among older adults and people with chronic illnesses [47,48,49]. However, concerns over the vaccine’s safety and side-effects are just as realistic [49,50], and may impede the progress of mass vaccination. The Center for Disease Control and Prevention [51] has recently relaxed guidelines about social distancing and mask use for people who been fully vaccinated. While the vaccinated individuals may enjoy the gradual resumption of their daily lives and routines, those who are yet to be vaccinated for health reasons may feel even more isolated, stigmatized and ‘left behind’ by the vaccinated crowd. Hence, support should be continued for those who are not vaccinated for health reasons, in order to ensure that disparity in COVID-19 vaccination will not generate a new category of inequality. In addition, after more than a year of upended lifestyles, it is foreseeable that the return to normality in full scale will take a ‘leap of faith’, especially for people with chronic illnesses. Thus, continual psychological support for managing stress and anxiety and fostering psychological resilience will be needed to help chronic illness patients adapt to the post-pandemic life.

## 5. Conclusions

Using a sample of participants with chronic illnesses in Hong Kong, we illustrated how changes in daily lifestyles may relate to poorer mental health and psychological resilience. Specifically, we found that the link between changes in daily lifestyles and mental health outcomes was mediated by psychological resilience. We recommend health and social care professionals dispense practical, up-to-date and feasible guidelines for supporting people with chronic illnesses to continue their daily routines while minimizing the infection risk, as well as adopt scalable and efficacious support to foster psychological resilience through tele-counselling or tele-medicine.

## Figures and Tables

**Figure 1 ijerph-18-05875-f001:**
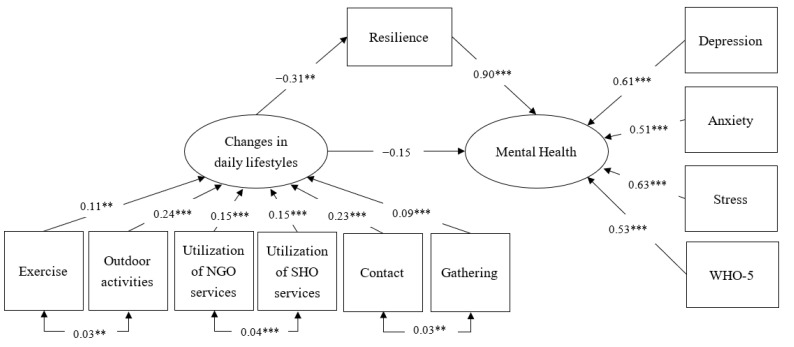
Results of SEM. Model fit statistics: Goodness of Fit Index (GFI) = 0.91; Root Mean Square Error of Approximation (RMSEA) = 0.09 (90% confidence interval = 0.08–0.10); Standardized Root Mean Square Residual (SRMR) = 0.08. ** *p*-value < 0.01; *** *p* < 0.001.

**Table 1 ijerph-18-05875-t001:** Respondents’ demographics (*N* = 408).

Variables	*N*	%
Age		
<55	106	26.0%
55–64	145	35.5%
65 or above	157	38.5%
Sex		
Female	250	61.3%
Male	158	38.7%
Type of chronic disease		
Hypertension	167	40.9%
Diabetes	123	30.1%
Heart disease	61	15.0%
Rheumatoid arthritis	47	11.5%
Stroke	37	9.1%
Systemic Lupus Erythematosus	20	4.9%
Ankylosing Spondylitis	28	6.9%
Cancer	17	4.2%
Asthma	11	2.7%
Brain injury	6	1.5%
Epilepsy	6	1.5%
Disabling effect of chronic conditions		
1 type	243	59.6%
2 types	107	26.2%
3 types	44	10.8%
4 types	14	3.4%
Working status		
Yes (full-time/part-time/self-employment)	126	30.9%
No	282	69.1%
Perceived risk of infection of COVID-19 (Mean, SD)	3.79	2.12

**Table 2 ijerph-18-05875-t002:** Descriptive statistics and correlations among outcome variables (*N* = 408).

Variables	Mean	SD	Bivariate Correlations			
DASS-21			2	3	4	5
1. Depression	7.64	8.47	0.62	0.73	−0.59	−0.54
Mild	42	10.3%				
Moderate	60	14.5%				
Severe or above	47	11.5%				
2. Anxiety	8.02	7.35		0.62	−0.45	−0.41
Mild	42	10.3%				
Moderate	60	14.7%				
Severe or above	47	11.5%				
3. Stress	10.93	9.08			−0.61	−0.57
Mild	40	9.8%				
Moderate	55	13.5%				
Severe or above	27	6.6%				
4. WHO-5	47.66	22.44				0.64
With depressive symptoms	215	52.7%				
5. CDRS-10	26.99	7.86				

Note: All bivariate correlations were significant at *p* < 0.001. Abbreviations: DASS-21, 21-item Depression, Anxiety and Stress Scale; WHO-5, World Health Organisation- Five Well-Being Index; CDRS-10, 10-item Connor–Davidson Resilience Scale.

**Table 3 ijerph-18-05875-t003:** Changes in daily lifestyles (*N* = 408).

Daily Lifestyles	Increase	Decrease	Same	None	Change or None
	*N* (%)				
Physical exercises	55 (13.5%)	189 (46.3%)	143 (35.0%)	21 (5.1%)	265 (65.0%)
Outdoor activities	16 (3.9%)	278 (68.1%)	106 (26.0%)	8 (2.0%)	302 (74.0%)
Utilization of NGO services	22 (5.4%)	256 (62.7%)	53 (13.0%)	77 (18.9%)	355 (87.0%)
Utilization of SHO services	17 (4.2%)	200 (49.0%)	42 (10.3%)	149 (36.5%)	366 (89.7%)
Contact with others	41 (10.0%)	202 (49.5%)	154 (37.7%)	11 (2.7%)	254 (62.3%)
Gathering with others	3 (0.7%)	363 (89.0%)	33 (8.1%)	9 (2.2%)	375 (91.9%)

Abbreviations: NGO, Non-government Organization; SHO, Self-help Organization.

**Table 4 ijerph-18-05875-t004:** Relationships between changes in daily lifestyles and psychological resilience and mental health (*N* = 408).

Change or None of Life Behavior	No		Yes		*p*-Value
	Mean	SD	Mean	SD	
Exercise					
DASS-21 Depression	6.02	7.97	8.52	8.61	0.004
DASS-21 Anxiety	6.76	7.14	8.70	7.38	0.010
DASS-21 Stress	8.74	8.92	12.11	8.97	0.000
WHO-5	51.05	22.86	45.83	22.04	0.027
CDRS-10	28.36	7.48	26.25	7.98	0.008
Outdoor Activities					
DASS-21 Depression	6.31	8.63	8.11	8.38	0.063
DASS-21 Anxiety	6.87	7.42	8.43	7.29	0.063
DASS-21 Stress	8.92	9.03	11.64	9.01	0.008
WHO-5	53.25	22.60	45.70	22.09	0.003
CDRS-10	29.11	7.90	26.24	7.72	0.001
Utilization of NGO services					
DASS-21 Depression	6.27	7.84	7.85	8.55	0.182
DASS-21 Anxiety	6.34	7.27	8.27	7.34	0.076
DASS-21 Stress	9.58	9.03	11.13	9.09	0.247
WHO-5	53.89	22.87	46.73	22.26	0.037
CDRS-10	28.02	8.12	26.83	7.82	0.322
Utilization of SHO services					
DASS-21 Depression	6.08	8.04	7.82	8.51	0.193
DASS-21 Anxiety	5.75	6.78	8.28	7.38	0.027
DASS-21 Stress	7.67	8.08	11.30	9.13	0.009
WHO-5	57.14	22.41	46.57	22.22	0.006
CDRS-10	29.74	6.95	26.67	7.91	0.010
Contact with others					
DASS-21 Depression	6.00	8.18	8.64	8.50	0.002
DASS-21 Anxiety	6.75	6.94	8.79	7.49	0.006
DASS-21 Stress	8.48	8.64	12.42	9.04	0.000
WHO-5	54.05	21.16	43.78	22.35	0.000
CDRS-10	29.29	7.73	25.59	7.62	0.000
Gathering with others					
DASS-21 Depression	6.68	8.94	7.73	8.43	0.521
DASS-21 Anxiety	7.32	7.29	8.08	7.36	0.567
DASS-21 Stress	9.55	9.15	11.05	9.08	0.370
WHO-5	56.12	20.14	46.91	22.50	0.017
CDRS-10	29.33	8.78	26.78	7.75	0.115

Note: *p*-value derived from independent *t*-tests.

**Table 5 ijerph-18-05875-t005:** Results of mediation analysis (with bootstrapping).

Effect	Estimate	95% LCI	95% UCI	*p*-Value	%
Direct effect (changes in daily lifestyles → mental health)	−0.15	−0.36	0.05	0.147	35.2%
Indirect effect (changes in daily lifestyles → resilience → mental health)	−0.28	−0.44	−0.10	0.001	64.8%
Total effect	−0.43	−0.72	−0.12	0.004	

Abbreviations: LCI, Lower Confidence Interval; UCI, Upper Confidence Interval.

## Data Availability

The data of this study can be obtained by emailing the corresponding authors.

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
