# Peer review of "Resilience in the Storm: Impacts of Changed Daily Lifestyles on Mental Health in Persons with Chronic Illnesses under the COVID-19 Pandemic"

_ijerph, 2021, doi:10.3390/ijerph18115875_

Round 1

Reviewer 1 Report

Dear Authors,

the impact of the current pandemic is transversal to human nature itself, affecting health and well-being at all levels. 

Paper needs a finishing touch and some insights in order to raise the bar of a paper, at a first glance already a notch above the others.

Thank you for sharing your contribution proposal

  • Why not SARS-CoV-2 among keywords? why not i.e. stigma/discrimination, disparities, inequalities?
  • Please contextualize and deepen the current pandemic scenario when introducing your work;
  • you did not investigate the fear of being stigmatized or discriminated as HCW or simply if they test positive or a family member / close cohabitant tests positive for SARS-CoV-2. This is a limitation that you may consider and deal with;
  • conclusion must be improved, too poor. What are the possible repercussions? What suggestions to give to the health policy maker?
  • which criteria for the selection of psychometric tools? The General Health Questionnaire (GHQ) scale could have provided some more hints for your discussion;
  • deal with gender: even if not statistically significant, I think there may be some interesting conclusions;
  • you must consider we are facing a CoViD-19 n-surge and your results could be up to date, but you should deal this in the introduction with the latest data regarding the second flow and consider it into discussion and conclusion sections;
  • what about the risk perception?

Discussion's paragraphs can be improved further. 

You need conclusions, too port in this first attempt. What are the possible repercussions? What suggestions to give to the health policy maker? Define a clear "take home message" from your perspective and address a conclusion section.

  1. The alleged identification of "scientific" bases of stigmata characterizing certain groups of population being the first, decisive and irreversible step towards the creation of a sort of "expendable victims", according to one well known pattern from the history of this kind of human affairs. 
  2. You should also refer toother examples of scientific literature that have completely misled epidemiological findings

Please update these gaps referring to the following references:

  • Irigoyen-Camacho, M.E.; Velazquez-Alva, M.C.; Zepeda-Zepeda, M.A.; Cabrer-Rosales, M.F.; Lazarevich, I.; Castaño-Seiquer, A. Effect of Income Level and Perception of Susceptibility and Severity of COVID-19 on Stay-at-Home Preventive Behavior in a Group of Older Adults in Mexico City. Int. J. Environ. Res. Public Health 2020, 17, 7418
  • Baldassarre, A.; Giorgi, G.; Alessio, F.; Lulli, L.G.; Arcangeli, G.; Mucci, N. Stigma and Discrimination (SAD) at the Time of the SARS-CoV-2 Pandemic. Int. J. Environ. Res. Public Health 2020, 17, 6341
  • Sarah Dryhurst, Claudia R. Schneider, John Kerr, Alexandra L. J. Freeman, Gabriel Recchia, Anne Marthe van der Bles, David Spiegelhalter & Sander van der Linden (2020) Risk perceptions of COVID-19 around the world, Journal of Risk Research, DOI: 10.1080/13669877.2020.1758193
  • Wong, B.Y.-M.; Lam, T.-H.; Lai, A.Y.-K.; Wang, M.P.; Ho, S.-Y. Perceived Benefits and Harms of the COVID-19 Pandemic on Family Well-Being and Their Sociodemographic Disparities in Hong Kong: A Cross-Sectional Study. International Journal of Environmental Research and Public Health 2021, 18, 1217
  • Weinstein, B.; da Silva, A.R.; Kouzoukas, D.E.; Bose, T.; Kim, G.J.; Correa, P.A.; Pondugula, S.; Lee, Y.; Kim, J.; Carpenter, D.O. Precision Mapping of COVID-19 Vulnerable Locales by Epidemiological and Socioeconomic Risk Factors, Developed Using South Korean Data. International Journal of Environmental Research and Public Health 2021, 18, 604
  • Dye, T.D.; Alcantara, L.; Siddiqi, S.; Barbosu, M.; Sharma, S.; Panko, T.; Pressman, E. Risk of COVID-19-related bullying, harassment and stigma among healthcare workers: an analytical cross-sectional global study. BMJ Open 2020, 10, e046620

In the conclusions you should refer to the very recent introduction of vaccines; what would change in this scenario? deal with it, even because CoViD-19 vaccines are now available all over the globe

Your contribution has little importance from an epidemiological and biostatic point of view, but it could be crucial for equality and human rights, offering a timely suggestion to the health authorities and policy makers.

Reviewer 2 Report

Thank you for the opportunity to review this interesting manuscript. I have a few minor comments for the authors to consider:

  • Abstract:
    • Please include a sentence outlining the aim of the research (or the research question/research hypotheses).
    • Line 18: Please include the name of the measurement tool used for resilience.
    • Lines 21-22: Please also include the estimated coefficients and their 95% CIs to illustrate the strength of the associations.
  • Introduction:
    • Lines 140-151: The sentences could be moved elsewhere because they would fit better under the Method section, and some would fit better under the Discussion/Implication section of the manuscript.
  • Methods:
    • Section 2.1: Please include the inclusion/exclusion criteria of the participants in the last paragraph. For instance, were they excluded because they did not have a social media account, or could not respond to the surveys written in Chinese, etc? This information may aid the discussion around the strength and limitation of this study.
    • Section 2.2: “Changes in daily lifestyles: Please clarify if expert panel was involved in selection of the “common and critical factors” (line 202-203). Were the questions verified for face validity, content validity, internal consistency, test-retest reliability, etc? If so, please add information around this. If not, please add this as a limitation of the study in the discussion section.
    • Section 2.4: The authors stated in line 215-218 that “Associations among the changes in daily lifestyles, mental health and psychological resilience were explored with bivariate correlations and analyses of variance (ANOVAs) with post-hoc tests with Tukey’s Test.”. Whilst the scales were continuous, the meaning of these scales were actually categorical. Please include a rationale to assess them using ANOVAs, instead of Chi-squared tests, or revise the method and update the results section (Table 2) with the results of Chi-squared tests. Further analyses on the mediation effect could potentially be assessed using logistic regression model to aid the interpretation of the results for clinicians.
  • Results:
    • Please indicate how many people were sent the survey to allow calculation of the response rate. This would be the first sentence of the Results section.
    • Throughout the results section, please revise all sentences that began with numbers so that the sentences do not begin with numbers. For example, revise “408 participants…” to “Four hundred and eight participants…” instead.
    • Please consider update the results (Table 2), Section 3.1 and 3.2, or provide a rationale supported by literature to no update.
    • Please keep the terminology consistent for association (categorical outcome variables) and correlation (continuous outcome variables).
  • Discussion:
    • Line 314: “…, we reckon …”. Please change ‘reckon’ to another word, e.g. believe, consider, etc, for a scientific paper.
    • Line 329: “As most of our participants were young-olds…”, did the authors mean young, or old, or cover a wide range of age?
    • May add a comment to indicate if the sample of 408 participants was representative of the target population by referring to the distribution of the demographic variables.
    • The authors may add a few more limitations, such as limited by the cross-sectional study design, and lack information around the length of their chronic illnesses (those who lived with chronic illnesses for a longer period may have different resilience to those who were recently diagnosed), lack of data around their past pandemic experience (unknown if the participants also lived through the 2003 SARS or migrated recently), lack of data collected around internalised mental wellbeing training (e.g. self-care apps, personal meditation apps, etc). These limitations serve as useful guide for future studies.
    • Line 383-385: The authors stated that “Care professionals of different medical disciplines and social contexts should be sensitive to the specific needs of their patients.”. Whilst this is true, there were no results from this study that supported such statement, especially since there were no data collected in this context. Please remove.
  • Conclusions: Please revise the conclusions so that it only highlights the main findings (the strengths of the study would be noted in the discussion section, not in the conclusion), and the recommendations are based on the study findings, i.e. revise the last sentence to “Therefore, we call for support for the chronic illness populations to continue their daily life and sustain their psychological resilience.” 

Round 2

Reviewer 1 Report

Conclusions still need to be improved

Author Response

Thanks for the further comment.

We understand the recent introduction of vaccines will have substantial implications on the daily lives of chronic illness patients and therefore their mental health. To highlight the role of vaccination, we have relocated the lines from Conclusion to a specific paragraph located after the Study Limitations (See Line 427-443). Also, we emphasized that continual support should be provided to people who are not vaccinated for health reasons, and that a return to the post-pandemic 'normality' may require adjacent psychological support for the management of stress and anxiety, especially for people with chronic illnesses'. We agree with the reviewer that inequality, stigma and health disparities are key topics in COVID adjustment and that vaccination itself should not generate another category of inequality.